# Cherry on Top: Parameter Heterogeneity and Quantization in Large Language Models

**Wanyun Cui**[*†‡], **Qianle Wang**[*†]

[†]Shanghai University of Finance and Economics
[‡]MoE Key Laboratory of Interdisciplinary Research of Computation and Economics,
Shanghai University of Finance and Economics
cui.wanyun@sufe.edu.cn, wql20000111@stu.sufe.edu.cn

## Abstract

This paper reveals the phenomenon of parameter heterogeneity in large language models (LLMs). We find that a small subset of "cherry" parameters exhibit a disproportionately large influence on model performance, while the vast majority of parameters have minimal impact. This heterogeneity is found to be prevalent across different model families, scales, and types. Motivated by this observation, we propose CherryQ, a novel quantization method that unifies the optimization of mixed-precision parameters. CherryQ identifies and preserves the critical cherry parameters in high precision while aggressively quantizing the remaining parameters to low precision. Extensive experiments demonstrate the effectiveness of CherryQ. CherryQ outperforms existing quantization approaches in terms of perplexity and downstream task performance. Notably, our 3-bit quantized Vicuna-1.5 exhibits competitive performance compared to their 16-bit counterparts.

## 1 Introduction

The rapid development of large language models (LLMs) has increased the demand of efficient deployment in various environments [1, 2, 11, 23]. However, the parameter size poses significant challenges for GPU memory requirements. Quantization, which reduces the bit-width of model parameters, has emerged as a solution to alleviate memory constraints of LLM deployment [12, 14, 18, 25, 26].

Low-precision parameter representation leads to quantization errors. Surprisingly, existing research has shown that LLMs exhibit a high robustness for quantization errors even for low-bit settings. For example, although 4-bit quantization can only represent 16 distinct values, even the simplest round-to-nearest strategy does not significantly degrade performance [17]. This raises the question: *what causes LLMs to be robust to quantization*?

We explore the parameters and answer this question via **Parameter Heterogeneity**, which refers to the significant variation in the influence of quantization on different parameters. We reveal that for the vast majority ($> 99\%$) of normal parameters, the effect of their quantization to the model are minimal and can thus be alleviated or ignored. However, there exists a small subset ($< 1\%$) of "cherry" parameters for which the effect are substantial and hard to mitigate.

Consider Figure 1a as an example. We show a scatter plot of the impacts on the model loss when perturbing each individual parameter in a parameter matrix from LLaMA2-7b [23]. The derivation of impacts is detailed in § 3. While 99% of the parameters fall within the range of (0, 0.1), a small subset of "cherry" parameters exhibits values ranging from (5, 30), which is 50-300 times higher

---

[*]Equal contribution

38th Conference on Neural Information Processing Systems (NeurIPS 2024).

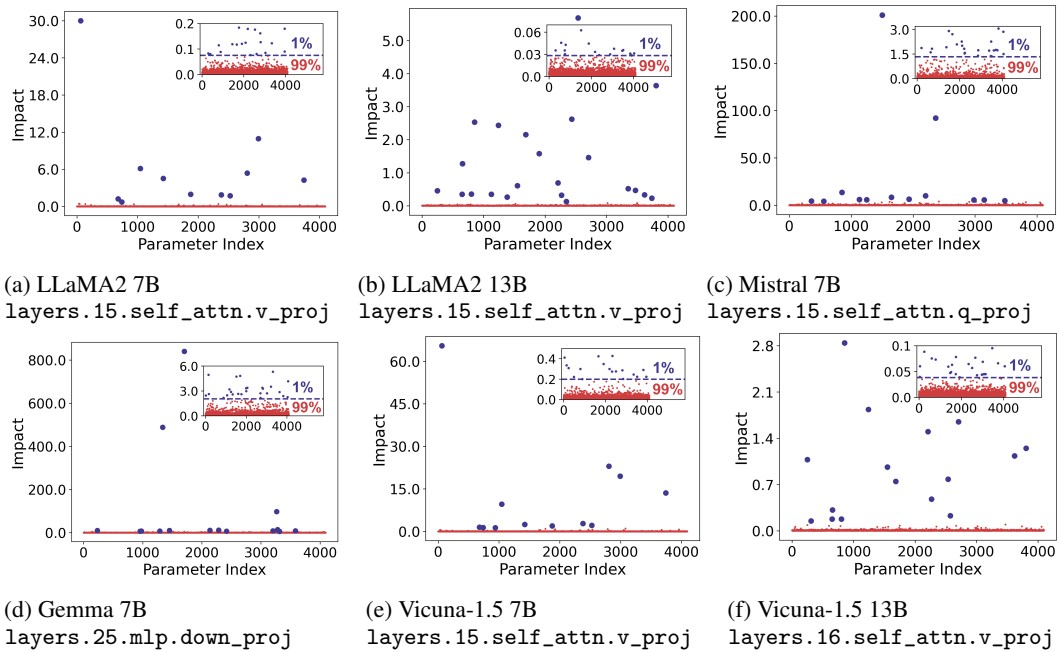

(a) LLaMA2 7B
`layers.15.self_attn.v_proj`

(b) LLaMA2 13B
`layers.15.self_attn.v_proj`

(c) Mistral 7B
`layers.15.self_attn.q_proj`

(d) Gemma 7B
`layers.25.mlp.down_proj`

(e) Vicuna-1.5 7B
`layers.15.self_attn.v_proj`

(f) Vicuna-1.5 13B
`layers.16.self_attn.v_proj`

Figure 1: Scatter plot of parameter impacts in different LLMs. We randomly sampled 4096 parameters from the corresponding parameter matrix. Each point represents the impact of an individual parameter. Insets show the zoomed-in y-axis. The heterogeneity is found across different model scales (1a,1b), different model families (1c, 1d), and both base models and chat models (1e, 1f).

than the *maximum* value of the remaining 99% of parameters. Moreover, this phenomenon is not an isolated case. We observed similar patterns across different scales of LLMs (Figure 1a1b), different families of LLMs, including Mistral [11] (Figure 1c) and Gemma [22] (Figure 1d), and both base models and chat models (Vicuna-1.5 [5] Figure 1e1f). The consistent presence of this phenomenon suggests that parameter heterogeneity is an inherent characteristic of LLMs.

Therefore, 99% of normal parameters explain the high robustness of LLMs to quantization errors. However, the small number of cherry parameters still leads to performance degradation under quantization. Consequently, the key to reducing quantization errors lies in addressing the quantization of cherry parameters.

The parameter heterogeneity also explains the previously discovered effectiveness of mixed-precision quantization strategies [6, 12, 17, 18]. By preserving a small proportion of parameters with high precision, the quantization performance can be effectively improved. Based on the heterogeneity, this strategy alleviates the impact of cherry parameters on model performance by maintaining their precision.

Indeed, mixed-precision strategies can effectively address the quantization error issue of cherry parameters. However, the core challenge lies in identifying cherry parameters based on specific metrics. Different metrics of parameter effects include weights [6, 13, 18], activations [17, 24], output changes [8], as well as the impact of parameters on model loss used in this paper. Based on the parameter heterogeneity, we argue that effective cherry parameter identification metrics should exhibit high heterogeneity, clearly distinguishing between cherry parameters and normal parameters. Accordingly, we compare three different metrics in Sec 5 and find that the impact best distinguishes cherry parameters from normal parameters. The experimental results in Sec 6.5 verify that choosing metrics with higher discriminative capability indeed leads to better performance.

Based on the above analysis, we design the CherryQ quantization algorithm, which selects cherry parameters based on the impact metric and end-to-end optimizes parameters with mixed precisions. Extensive experiments on various models and benchmarks demonstrate the effectiveness of CherryQ. It consistently yields the lowest perplexity in most settings. Notably, our 3-bit Vicuna-1.5 model

exhibits performance on par with the 16-bit counterpart on Vicuna-bench [5]. Our 2-bit quantization method significantly outperforms the SOTA approaches.

In summary, by systematically revealing parameter heterogeneity in LLMs, we answer the following questions:

1. *What causes the high robustness of LLMs to quantization?* It is due to the 99% of normal parameters in parameter heterogeneity.

2. *Why is mixed-precision quantization effective?* This strategy addresses the quantization problem of the very few cherry parameters in parameter heterogeneity.

3. *How to find the optimal mixed-parameter selection strategy?* Based on heterogeneity that distinguishes normal parameters from cherry parameters, the impact-based metric demonstrates the highest discriminative capability.

4. *How to quantize parameters according to parameter heterogeneity?* We propose CherryQ based on these findings. Extensive empirical results verify its effectiveness in 2-, 3-, 4-bit quantization.

## 2   Related Work

**Quantization Strategies for LLMs** Various quantization strategies have been proposed in the literature to reduce the precision of weights and activations while maintaining acceptable accuracy. These strategies can be broadly classified into post-training quantization and quantization-aware training [14]. Post-training quantization methods, such as OBD, OBS, and GPTQ, directly quantize the pre-trained model without fine-tuning [15, 10, 9]. On the other hand, quantization-aware training methods, such as LLM-QAT [18], incorporate quantization operations into the training process to jointly optimize the quantized model. Some works also explore mixed-precision quantization [12] and adaptive quantization bins [7] to achieve a better trade-off between accuracy and efficiency.

**Outliers in Language Model Quantization** The idea of modeling parameter outliers in LLM quantization is not new. Exploring outliers mainly includes the perspectives of magnitude [18, 7] and activations [4, 6]. For example, from the magnitude perspective, QLoRA assumes that parameters follow a Gaussian distribution [7] and designs information-theoretically optimal quantized bins based on this assumption. [18] keeps outlier parameters in 16-bit precision. From the activation perspective, [17] migrates the outlier amplifier to subsequent modules through an equivalent transformation. Additionally, SqueezeLLM also measures outliers from the perspective of parameter impact [12]. To the best of our knowledge, our work is the first to systematically reveal the outliers (heterogeneity) of parameter impact across different models, and we show a more pronounced imbalance in parameter impacts compared to magnitudes (§ 6.5). Furthermore, we propose a method to unify outlier (cherry) parameter optimization and normal parameter optimization, addressing the optimization challenges of heterogeneous parameters.

## 3   Quantifying the Impact of Parameters on Model Performance

The impact of parameters on model performance is quantified by the increase of the training loss when perturbing the parameter weight, which is widely used in post-training quantization approaches [15, 10, 9]. We adopt a second-order Taylor approximation of the training loss w.r.t. parameter perturbation. Given a parameter $w_i$ and a small perturbation $\Delta$ applied to it, such that $w_i \leftarrow w_i + \Delta$, the change in the training loss can be expressed as:

$$\mathbf{L}(w_i + \Delta) - \mathbf{L}(w_i) = g_i \Delta + \frac{1}{2}\mathbf{H}_{ii}\Delta^2 + O(\Delta^2) \tag{1}$$

where $g_i = \mathbb{E}[\frac{\partial L}{\partial w_i}]$ represents the expected gradient of the loss with respect to $w_i$, and $\mathbf{H}_{ii} = \mathbb{E}[\frac{\partial^2 L}{\partial w_i^2}]$ denotes the $i$-th value of the Hessian matrix of the loss. Since the target model is a well-converged model, we can assume that $g_i \approx 0$, simplifying the expression to:

$$\mathbf{L}(w_i + \Delta) - \mathbf{L}(w_i) \approx \frac{1}{2}\mathbf{H}_{ii}\Delta^2 \tag{2}$$

Therefore, $\mathbf{H}_{ii}$ quantify the impact of quantization-induced perturbations on the model's training loss. Parameters with larger values of $\mathbf{H}_{ii}$ exhibit higher sensitivity to quantization and require careful treatment to maintain model performance. We denote $\mathbf{H}_{ii}$ as the impact of $w_i$.

**Efficient Computation** Computing $\mathbf{H}_{ii}$ of the diagonal of Hessian matrix for each parameter is computationally expensive, particularly for large-scale models. To overcome this challenge, we propose an efficient approximation using the Fisher Information Matrix ($\mathbf{F}$). Since $\mathbf{H}$ is the Hessian matrix of a negative log-likelihood loss, $\mathbf{H}$ is equal to Fisher information matrix [16]. For the diagonal of the Hessian matrix, we have:

$$\mathbf{H}_{ii} = \mathbf{F}_{ii} = \mathbb{E}[g_i^2] \tag{3}$$

## 4 End-to-End Mixed-Precision Quantization

The insights gained from Figure 1 highlight the heterogeneity in model parameters. To mitigate the impact of cherry parameters on quantization, we propose to preserve their high-precision values during the quantization process. By maintaining the fidelity of these critical parameters, we ensure that the essential information they capture is not compromised.

Optimizing mixed-precision parameters in LLMs presents a unique challenge in the widely adopted Post-Training Quantization (PTQ) framework [14]. If we do not allow the updates of the cherry parameters, the quantization will certainly lose the flexibility provided by these critical parameters. This prevents the cherry parameters from reaching their optimum. On the other hand, PTQ struggles to simultaneously optimize high-precision cherry parameters and low-precision normal parameters. This is because the cherry parameter updates during the PTQ process significantly affect the optimal values of the normal parameters. So normal parameters need to be continually updated as the cherry parameter varies. However, in PTQ, once the normal parameters are quantized, they cannot be further updated. This prevents the early-stage quantized parameters from reaching their optimal values.

To address this challenge, we propose a novel approach that end-to-end optimize the mixed-precision parameters via backpropagation. Our method leverages a quantization-aware training framework. To simultaneously optimize both the cherry parameters and normal parameters, we use two separate backpropagation strategies. The high-precision cherry parameters are updated using standard gradient descent, while the low-precision normal parameters employ the Straight-Through Estimator (STE) trick [3] for low-precision gradient descent. This unified backpropagation enables the end-to-end optimization of both cherry parameters and normal parameters, enhancing the overall optimization effect. We show the quantization in Algorithm 1.

---

**Algorithm 1** CherryQ

**Require:** Model parameters $\mathbf{W}$, quantization function $Quant(\cdot)$, threshold $\tau$, learning rate $\eta$
**Ensure:** Quantized model parameters
1: $\mathbf{C} \leftarrow \{w_i \in \mathbf{W} \mid \mathbf{H}_{ii} > \tau\}$                    ▷ Identify cherry parameters
2: $\mathbf{N} \leftarrow \mathbf{W} \setminus \mathbf{C}$                    ▷ Identify normal parameters
3: **for** each training batch $x$ **do**
4:     $L \leftarrow \text{model}(x; \mathbf{C} \cup Quant(\mathbf{N}))$          ▷ Compute loss w.r.t. mixed-precision parameters
5:     $\mathbf{C} \leftarrow \mathbf{C} - \eta \frac{\partial L}{\partial \mathbf{C}}$                    ▷ Standard gradient descent for cherry parameters
6:     $\mathbf{N} \leftarrow \mathbf{N} - \eta \cdot \text{STE}(\frac{\partial L}{\partial \mathbf{N}})$          ▷ Gradient approximation by STE for normal parameters
7: **end for**
8: **return** $\mathbf{C} \cup Quant(\mathbf{N})$

---

## 5 Heterogeneity-based Cherry Parameter Selection

Correctly identifying cherry parameters is one of the main challenges of CherryQ quantization. Candidate metrics for parameter influences include weights [13, 18, 6], activations [17, 24], and impacts ($\mathbf{H}_{ii}$). We propose that an effective metric should reflect heterogeneity, specifically by differentiating the influence of cherry parameters and normal parameters of the model.

To this end, we define the heterogeneity score. In Figure 1, a small subset of parameters exhibit significantly higher impacts compared to the maximum of the majority. Inspired by this, the heterogeneity

score is defined as the ratio of the *mean* impact of the top 1% parameters to the *maximum* impact of the bottom 99% parameters, as shown in Equation (4). A higher heterogeneity score indicates a more significant disparity in parameter importance.

$$\text{Heterogeneity Score}(f) = \frac{\text{Mean}(f(w_i)_{\text{top 1\%}})}{\text{Max}(f(w_i)_{\text{bottom 99\%}})} \tag{4}$$

where $f(w_i)$ denotes the parameter influence for parameter $w_i$, and $f$ is chosen from parameter weights, activations, and impacts.

Figure 2 presents the heterogeneity scores for different metrics across various LLMs. The impact-based metric consistently shows higher heterogeneity scores compared to weights and activations. This indicates that the impact metric better distinguishes between the normal and cherry parameters, thus providing a more effective means of identifying cherry parameters. The validity of using heterogeneity scores for cherry parameter selection will be further verified in Sec 6.5, demonstrating that higher heterogeneity scores lead to better model performance.

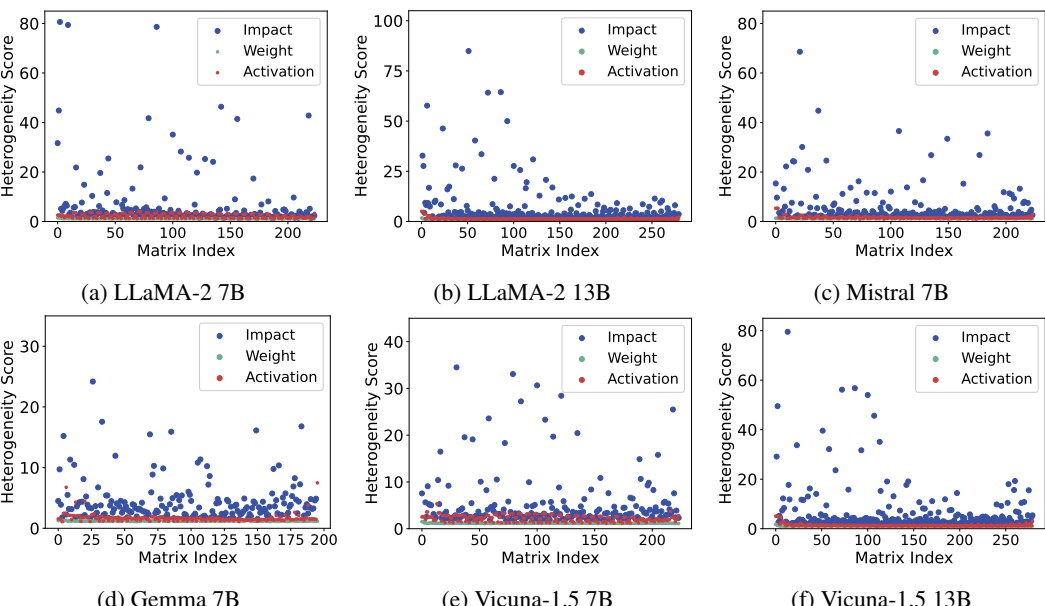

Figure 2: Scatter distribution of heterogeneity scores for different parameter matrices in LLMs. Each point represents a parameter matrix.

**Data Independence** We investigate whether impact-based parameter heterogeneity exhibits data independence - that is, whether different data samples share the same cherry parameters. Applying identical cherry parameters across different samples is only valid when there is data independence. To examine data independence within the same dataset, we randomly selected five sets of 128 WikiText-2 samples. We evaluated the overlap of their cherry parameters in each pair of sample sets. To evaluate cross-dataset data independence, we performed 9 independent sampling trials, each collecting 128 samples from C4 and 128 samples from WikiText-2 to evaluate the overlap. For Vicuna, we further added Sharegpt as a data source. Table 1 presents the overlap ratios of the cherry parameters. Despite the fact that cherry parameters constitute only $1/256$ of the total parameters, all models demonstrate significant overlap ratios. This finding suggests that the cherry parameters possess an inherent data-independent nature.

# 6 Quantization Experiments

## 6.1 Implementation Details

**Parameter Representation:** Based on the observation that cherry parameters occupy a very small proportion, for each row of parameters in each parameter matrix, we consider only the top $1/256$

Table 1: Overlap ratio (%) of cherry parameters (top 1/256), where L2: LLaMA-2, V: Vicuna-1.5, M: Mistral, G: Gemma.

| Model | L2-7B | L2-13B | V-7B | V-13B | M-7B | G-7B |
|---|---|---|---|---|---|---|
| Within dataset | 84 | 75 | 68 | 63 | 89 | 90 |
| Across datasets | 68 | 66 | 65 | 60 | 85 | 86 |

parameters with the highest impact as cherry parameters and retain their FP16 precisions. For example, the parameter matrix size of LLaMA2-7B is $4096 \times 4096$. So we average the impact across all rows for each column and then select the top 16 columns with the highest average impact, resulting in $16 \times 4096$ parameters as cherry parameters. Furthermore, to recover the complete parameter matrix, an INT16 is required to record the indices of these 16 columns. Thus, the storage overhead for the column indices is minimal. For normal parameters, we employ *full range symmetric MinMax quantization* to quantize their weights. And we adopt a widely-used parameter grouping strategy. For more details, see Sec C.

**Quantization Datasets:** For the quantization of the base LLMs, we follow [9] to use C4 [20] as the training data. We selected the first four partitions of C4 and chose data with a length of $\geq 2048$ tokens, resulting in a total of 50k samples of 2048 tokens. For the chat LLMs, since Vicuna-1.5 [5] is obtained by supervised fine-tuning based on ShareGPT [5], we also use the ShareGPT dataset for training. We used a total of 20k training samples from ShareGPT for QAT and Cherry.

**Baselines** We compare our method with various quantization methods, including QAT [18], GPTQ [9], SqueezeLLM [12], OmniQuant [21], and AWQ [17]. For OmniQuant and AWQ, we use their results reported in [21]. For SqueezeLLM, we use the results in its original paper [12]. For GPTQ, its 4-bit model is obtained from the open-source [2]. Due to the lack of a 3-bit GPTQ model, we quantize the model ourselves via the implementation of Auto-GPTQ [3]. Since CherryQ is based on QAT, for fair comparisons, the implementation of QAT is the same as CherryQ, except that it does not handle cherry parameters.

## 6.2 Effect of Base LLM Quantization

In this section, we present the main experimental results to demonstrate the effectiveness of CherryQ on LLaMA2 [23]. We evaluate CherryQ with both perplexity and downstream tasks, comparing its performance with state-of-the-art quantization methods.

### 6.2.1 Perplexity Results

We follow [9, 21] to evaluate the perplexity of CherryQ on two widely used corpora: C4 and WikiText-2 [19]. We use the validation split of C4 to avoid data leakage. We show the results of 3-bit quantization using different quantization approaches in Table 2. We show the results of different model scales and different group sizes.

From the results, CherryQ consistently outperforms all other approaches across both model sizes (7B and 13B) and grouping sizes (64 and 128), achieving the lowest perplexity on both the C4 and WikiText-2 datasets. Notably, CherryQ's perplexity is significantly closer to the full-precision (FP16) baseline compared to other methods, highlighting its ability to preserve model performance after quantization.

Table 3 compares different 4-bit quantization methods. Again, CherryQ achieves the lowest perplexity scores in most settings, demonstrating its effectiveness in higher-bit quantization settings.

### 6.2.2 Downstream Task Performance

To further validate the effectiveness on specific tasks, we evaluated the quantized models on various downstream tasks from the HuggingFace OpenLLM Leaderboard. Table 4 presents the performance comparison of different 3-bit quantization methods for LLaMA2. **CherryQ consistently outper-**

---
[2]https://huggingface.co/TheBloke
[3]https://github.com/AutoGPTQ/AutoGPTQ

**forms other methods across almost all tasks**, achieving the highest average score. This showcases CherryQ's ability to maintain the model's generalization capabilities for downstream tasks.

Table 2: Perplexity (↓) of 3-bit quantization on LLaMA2 models. gX means the group size is X. The results of OmniQuant and AWQ are from [21]. The results of SqueezeLLM are from [12].

| Method | Avg. bit | 7B-3bit-g128 | | Avg. bit | 7B-3bit-g64 | | Avg. bit | 13B-3bit-g128 | | Avg. bit | 13B-3bit-g64 | |
|---|---|---|---|---|---|---|---|---|---|---|---|---|
| | | c4 | wiki2 | | c4 | wiki2 | | c4 | wiki2 | | c4 | wiki2 |
| FP16 | 16 | 6.97 | 5.47 | 16 | 6.97 | 5.47 | 16 | 6.47 | 4.88 | 16 | 6.47 | 4.88 |
| QAT | 3.13 | 9.25 | 6.90 | 3.25 | 8.74 | 7.13 | 3.13 | 7.19 | 5.63 | 3.25 | 7.02 | 5.48 |
| GPTQ | 3.15 | 8.28 | 6.74 | 3.30 | 8.20 | 6.62 | 3.15 | 7.24 | 5.63 | 3.30 | 7.10 | 5.56 |
| AWQ | 3.15 | 7.84 | 6.24 | - | - | - | 3.15 | 6.94 | 5.32 | - | - | - |
| OmniQuant | 3.15 | 7.75 | 6.03 | - | - | - | 3.15 | 6.98 | 5.28 | - | - | - |
| SqueezeLLM | - | - | - | 3.24 | 7.51 | 5.96 | - | - | - | 3.24 | 6.82 | 5.23 |
| **CherryQ** | 3.17 | **7.39** | **5.93** | 3.30 | **7.34** | **5.87** | 3.17 | **6.80** | **5.26** | 3.29 | **6.76** | **5.21** |

Table 5 extends the comparison to 4-bit quantization. CherryQ continues to excel, achieving the highest scores on most individual tasks and the highest average score overall. These results highlight the generalization ability of CherryQ across different quantization bits and model sizes.

Table 3: Perplexity (↓) of 4-bit quantization on LLaMA2 models.

| Method | Avg. bit | 7B-4bit-g128 | | Avg. bit | 13B-4bit-g128 | |
|---|---|---|---|---|---|---|
| | | c4 | wiki2 | | c4 | wiki2 |
| FP16 | 16 | 6.97 | 5.47 | 16 | 6.47 | 4.88 |
| QAT | 4.13 | 7.29 | 5.81 | 4.13 | 6.67 | 5.12 |
| GPTQ | 4.15 | 7.30 | 5.73 | 4.15 | 6.63 | 4.97 |
| AWQ | 4.15 | 7.13 | 5.62 | 4.15 | 6.56 | 4.97 |
| OmniQuant | 4.15 | 7.12 | 5.58 | 4.15 | 6.56 | **4.95** |
| **CherryQ** | 4.17 | **7.07** | **5.58** | 4.16 | **6.56** | 4.99 |

## 6.3 Effect of Chat LLM Quantization

We conducted experiments on Vicuna-1.5 [5]. We apply 3-bit quantization with group size=128 for CherryQ and other baselines.

**Evaluation** To assess the performance of quantized open-ended chat models, we employ a pairwise comparison on the Vicuna-bench [27], which consists of 80 test samples. We compare the responses generated by the quantized models against those generated by the original 16-bit Vicuna-1.5. The evaluation is performed using GPT-4, which automatically classifies the quantized model's response as "win", "tie", or "lose" relative to the FP16 model's response. To get rid of the ordering effect of the evaluation, we follow [17] to compare the responses with both orders, resulting in 160 trials.

Figure 3 presents the results of the pairwise comparison for each quantized model against its FP16 counterpart. The results demonstrate that CherryQ consistently outperforms other quantization baselines in preserving the performance of chat models. It achieves the highest number of wins and ties against the FP16 models, while minimizing the number of losses.

Notably, **3-bit CherryQ achieves a slightly better win-tie-lose ratio over the FP16 Vicuna model**, indicating that the 3-bit quantized model performs on par with or even better than the FP16 model. As intuitively CherryQ cannot surpass the target 16 bit model, we think the result suggests that CherryQ maintains almost all its performance even at 3 bit, making GPT-4 hard to distinguish the quality of low-bit and FP16 models.

## 6.4 Extreme 2-Bit Quantization

We further explore the extreme case of 2-bit quantization. Although 2-bit quantization greatly reduces memory requirements for model storage and inference, existing methods still show a significant performance gap compared to their 16-bit counterparts.

Table 4: Performance of different 3-bit quantization methods on Huggingface OpenLLM for LLaMA2-7B and LLaMA2-13B.

| Method | Hellaswag | Winogrande | ARC | TruthfulQA | GSM8K | MMLU | Average (↑) |
|--------|-----------|------------|-----|------------|-------|------|-------------|
| LLaMA2-7B-3bit-g64 | | | | | | | |
| FP16 | 78.6 | 74.0 | 53.2 | 38.8 | 14.5 | 46.7 | 51.0 |
| QAT | 75.5 | 71.6 | 49.2 | 37.3 | 7.3 | 40.6 | 46.9 |
| GPTQ | 73.9 | 71.7 | 48.6 | 38.8 | 8.1 | 39.4 | 46.8 |
| **CherryQ** | **77.0** | **71.8** | **50.6** | **38.6** | **10.4** | **43.9** | **48.7** |
| LLaMA2-7B-3bit-g128 | | | | | | | |
| FP16 | 78.6 | 74.0 | 53.2 | 38.8 | 14.5 | 46.7 | 51.0 |
| QAT | 75.4 | 70.8 | 48.2 | 37.7 | 6.7 | 39.0 | 46.3 |
| GPTQ | 72.9 | 70.8 | 48.6 | 39.1 | 5.4 | 38.2 | 45.8 |
| **CherryQ** | **76.3** | **72.4** | **49.7** | **38.1** | **8.8** | **41.6** | **47.8** |
| LLaMA2-13B-3bit-g64 | | | | | | | |
| FP16 | 82.1 | 76.6 | 59.4 | 37.4 | 22.5 | 55.7 | 55.6 |
| QAT | 80.7 | 75.1 | 55.5 | **39.0** | 16.8 | 52.9 | 53.3 |
| GPTQ | 79.2 | 74.4 | 56.5 | 36.0 | 16.4 | 52.4 | 52.5 |
| **CherryQ** | **81.1** | **76.2** | **57.3** | 38.0 | **18.4** | **53.5** | **54.1** |
| LLaMA2-13B-3bit-g128 | | | | | | | |
| FP16 | 82.1 | 76.6 | 59.4 | 37.4 | 22.5 | 55.7 | 55.6 |
| QAT | 80.7 | **75.5** | 55.3 | 38.8 | 16.0 | 51.9 | 53.0 |
| GPTQ | 79.1 | 75.4 | 54.1 | 34.9 | 15.6 | 50.3 | 51.6 |
| **CherryQ** | **81.0** | 75.4 | **56.7** | **38.9** | **17.8** | **52.5** | **53.7** |

Table 5: Performance comparison of different 4-bit quantization methods for LLaMA2-7B and LLaMA2-13B models over Huggingface OpenLLM Leaderboard.

| Method | Hellaswag | Winogrande | ARC | TruthfulQA | GSM8K | MMLU | Average (↑) |
|--------|-----------|------------|-----|------------|-------|------|-------------|
| LLaMA2-7B-4bit-g128 | | | | | | | |
| FP16 | 78.6 | 74.0 | 53.2 | 38.8 | 14.5 | 46.7 | 51.0 |
| QAT | 77.5 | 72.2 | **52.0** | 39.0 | 10.6 | 43.7 | 49.2 |
| GPTQ | 77.6 | 72.9 | **52.0** | 39.1 | 11.1 | 43.8 | 49.4 |
| **CherryQ** | **77.8** | **73.5** | 51.5 | **39.5** | **12.9** | **44.4** | **49.9** |
| LLaMA2-13B-4bit-g128 | | | | | | | |
| FP16 | 82.1 | 76.6 | 59.4 | 37.4 | 22.5 | 55.7 | 55.6 |
| QAT | 81.9 | 75.7 | 57.9 | **38.9** | 19.6 | 54.2 | 54.7 |
| GPTQ | 81.5 | 76.8 | 57.4 | 36.1 | 20.4 | 54.6 | 54.5 |
| **CherryQ** | **82.0** | **77.0** | **58.6** | 38.8 | **21.0** | 54.6 | **55.3** |

**Implementation Details** To achieve high-quality 2-bit quantization, we integrated the scaling-up trick introduced in [17]. Specifically, after identifying cherry and normal parameters, we automatically search for the optimal scale of each column of normal parameters that minimizes the output difference after quantization for each layer. The quantization function is formulated as $Q'(w) = Q(w \cdot s)/s$, where $Q(\cdot)$ represents standard asymmetric quantization on the min-max grid, and $s$ is a constant that scales up the normal parameters and remains fixed during the training process. The cherry parameters are excluded from quantization and retain their 16-bit precision throughout the grid search.

**Results** Table 6 presents the perplexities of 2-bit quantization on LLAMA2 models. Compared to existing methods such as GPTQ, AWQ, and OmniQuant, our proposed CherryQ method demonstrates superior performance across all metrics. Specifically, CherryQ achieves perplexity scores of 9.55

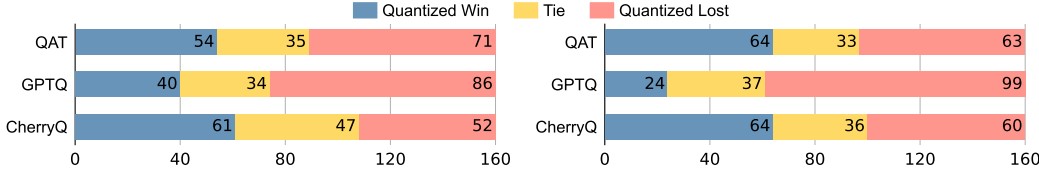

Figure 3: Comparison of 3-bit quantized models to FP16 Vicuna-1.5. (Left) Comparisons to Vicuna-1.5-7B. (Right) Comparisons to Vicuna-1.5-13B. CherryQ even shows competitive quality compared to the 16-bit counterpart.

Table 6: Perplexity (↓) of 2-bit quantization on LLaMA2 models. The results of GPTQ, AWQ and OmniQuant are from [21].

| Method | Avg. bit | 7B-2bit-g128 | | Avg. bit | 7B-2bit-g64 | | Avg. bit | 13B-2bit-g128 | | Avg. bit | 13B-2bit-g64 | |
|---|---|---|---|---|---|---|---|---|---|---|---|---|
| | | c4 | wiki2 | | c4 | wiki2 | | c4 | wiki2 | | c4 | wiki2 |
| FP16 | 16 | 6.97 | 5.47 | 16 | 6.97 | 5.47 | 16 | 6.47 | 4.88 | 16 | 6.47 | 4.88 |
| GPTQ | 2.15 | 33.70 | 36.77 | 2.30 | 19.40 | 20.85 | 2.15 | 20.97 | 28.14 | 2.30 | 12.48 | 22.44 |
| AWQ | 2.15 | $> 10^5$ | $> 10^5$ | 2.30 | $> 10^5$ | $> 10^5$ | 2.15 | $> 10^4$ | $> 10^5$ | 2.30 | $> 10^4$ | $> 10^5$ |
| OmniQuant | 2.15 | 15.02 | 11.06 | 2.30 | 12.72 | 9.62 | 2.15 | 11.05 | 8.26 | 2.30 | 10.05 | 7.56 |
| **CherryQ** | 2.19 | **9.55** | **8.34** | 2.34 | **9.08** | **7.84** | 2.19 | **8.40** | **7.20** | 2.33 | **8.02** | **6.72** |

and 8.34 in the 7B-3bit-g128 and 7B-3bit-g64 settings, respectively. These results significantly outperform other methods, validating the effectiveness of CherryQ in 2-bit quantization.

## 6.5 Comparison of Parameter Selection Criteria

To evaluate the effectiveness of our proposed impact-based parameter selection criterion, we conducted experiments comparing it with the criterions of parameter weights and activations. Table 7 presents the perplexity of LLaMA2-7B-3bit and LLaMA2-13B-3bit models, using both criteria for cherry parameter selection.

From the results, it is evident that the impact-based criterion consistently outperforms other criterions across all settings. These results demonstrate that our proposed impact-based criterion is a more effective measure to identify cherry parameters. The impacts identify and preserve the most critical parameters during the quantization process. These results are consistent with the analysis in Sec 5 regarding the effectiveness of heterogeneity scores in selecting cherry parameters.

Table 7: Perplexity (↓) of different parameter selection criteria.

| Method | LLaMA2-7B-3bit | | LLaMA2-13B-3bit | |
|---|---|---|---|---|
| | c4 | wiki2 | c4 | wiki2 |
| Weight-g64 | 7.93 | 6.40 | 6.91 | 5.35 |
| Activation-g64 | 7.37 | 5.89 | 6.77 | 5.22 |
| **Impact-g64** | **7.34** | **5.87** | **6.76** | **5.21** |
| Weight-g128 | 8.12 | 6.58 | 6.94 | 5.37 |
| Activation-g128 | 7.51 | 6.03 | 6.81 | 5.27 |
| **Impact-g128** | **7.39** | **5.93** | **6.80** | **5.26** |
| | LLaMA2-7B-4bit | | LLaMA2-13B-4bit | |
| Weight-g128 | 7.19 | 5.68 | 6.62 | 5.05 |
| Activation-g128 | 7.09 | 5.59 | 6.56 | 5.00 |
| **Impact-g128** | **7.07** | **5.58** | **6.56** | **4.99** |

# 7    Conclusion

In this paper, we systematically investigated the phenomenon of parameter heterogeneity in large language models (LLMs). Our experiments on LLaMA2, Mistral, Gemma, and Vicuna models consistently demonstrated that a small subset of parameters, referred to as "cherry" parameters, play a crucial role in maintaining the model's performance, while the vast majority of parameters can be quantized to ultra-low precision without significant degradation. This finding highlights the potential of the heterogeneous nature of parameter importance.

Motivated by this observation, we proposed a novel CherryQ quantization algorithm, which uses a quantization-aware training framework for the end-to-end optimization of both cherry parameters and normal parameters. Extensive experiments demonstrate that CherryQ achieves significantly lower perplexity scores and better downstream performance.

# 8    Limitations

There are some limitations to consider. First, the method relies heavily on the accurate identification of cherry parameters, which may vary across different model architectures and training datasets. This dependency could potentially limit the generalization ability of CherryQ to new or unseen models. Second, the computational overhead required for the impact-based identification and scaling of parameters, although justified by the performance gains, may pose challenges for extremely large models or those deployed in real-time systems with stringent latency requirements.

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

# A Effect of Chat LLM Quantization on MMLU

We further evaluate the performance of CherryQ on the MMLU benchmark by quantizing the Vicuna1.5 model. As shown in Table 8, CherryQ outperforms both QAT and GPTQ in terms of average accuracy across almost all categories.

Table 8: Comparison of different 3-bit quantization methods on zero-shot MMLU accuracy applied to Vicuna-1.5.

| Method | Humanities | STEM | Social Sciences | Other | Average |
|---|---|---|---|---|---|
| | | Vicuna1.5-7B-3bit-g128 | | | |
| FP16 | 46.8 | 39.4 | 57.9 | 56.3 | 49.9 |
| QAT | 43.4 | **37.7** | 53.0 | 52.4 | 46.4 |
| GPTQ | 42.7 | 37.3 | 53.0 | 51.0 | 45.7 |
| **CherryQ** | **43.8** | 37.2 | **54.3** | **53.5** | **46.9** |
| | | Vicuna1.5-13B-3bit-g128 | | | |
| FP16 | 50.2 | 43.5 | 63.0 | 62.0 | 54.3 |
| QAT | 47.8 | 40.1 | 58.6 | 58.1 | 50.9 |
| GPTQ | 46.1 | 39.4 | 57.6 | 55.2 | 49.3 |
| **CherryQ** | **49.0** | **40.6** | **60.2** | **58.8** | **51.9** |

# B Training Details

For all LLM scales (7B, 13B), and both base models and chat models (LLaMA2, Vicuna-v1.5), we train the models on a single node with 8 x A100 80GiB GPUs. We use a total batch size of 128, a learning rate of 2e-5, a weight decay of 0.0, a cosine scheduler with 5% warm-up steps. The final learning rate is 25% of the peak learning rate for 2/3-bit LLMs, 10% for 4-bit LLMs. We train 1 epoch on base models, 2 epochs on chat models.

# C Parameter Representation Details

For normal parameters, we employ *full range symmetric MinMax quantization* to quantize their weights [14]. Specifically, an FP16 value is mapped to the range of $[-2^{k-1}, 2^{k-1} - 1]$ and symmetrically distributed on both sides of the coordinate axis. The quantization of an FP16 tensor $X^{FP16}$ to $k$ bits is computed by:

$$X^{Intk} = \lfloor Clip(\frac{X^{FP16}}{S}, -2^{k-1} + \epsilon, 2^{k-1} - \epsilon) - 0.5 \rceil \tag{5}$$

where $\lfloor \cdot \rceil$ denotes the round function, $S$ is the quantization scaling factor $S = \frac{Max(|X^{FP16}|)}{2^{k-1}}$, and $\epsilon$ is a very small positive number (= 0.01 in our setting) to ensure that $X^{Intk}$ falls into the target range.

Dequantization restores the quantized integer values based on the scaling factor:

$$Dequant(S, X^{Intk}) = S(X^{Intk} + 0.5) \tag{6}$$

To further improve the quantization accuracy, we adopt a widely-used parameter grouping strategy. Specifically, the parameters are divided into groups in order, and each group independently calculates its scaling factor. For example, if we divide a parameter matrix $W \in \mathbb{R}^{r \times c}$ that needs to be quantized with a group size of $B$, we will obtain a total of $r \times (c/B)$ groups.

# D Licenses for Existing Assets

We list the assets used in this paper and their licenses below:

- [5], llama2
- https://huggingface.co/TheBloke, llama2
- https://github.com/AutoGPTQ/AutoGPTQ, MIT License
- [23], arXiv.org perpetual, non-exclusive license 1.0
- [18], arXiv.org perpetual, non-exclusive license 1.0
- [20], arXiv.org perpetual, non-exclusive license 1.0
- [19], arXiv.org perpetual, non-exclusive license 1.0

