# OpenReview forum: "Cherry on Top: Parameter Heterogeneity and Quantization in Large Language Models"
_NeurIPS.cc/2024/Conference — NeurIPS 2024 poster_

### Official Review · Reviewer_P3Ag · 2024-07-08

**Soundness:** 2
**Presentation:** 3
**Contribution:** 3
**Rating:** 5
**Confidence:** 4

**Summary:**

The paper introduces a quantization technique for LLMs, inspired by an empirical study on parameter importance. The authors identify a small subset of parameters, termed "Cherry parameters," which exhibit significantly higher importance than others. They propose an optimization strategy: retaining full precision for Cherry parameters and their gradients, while quantizing the normal parameters. A straight-through estimator is used to handle the gradients of these quantized parameters during optimization.

**Strengths:**

1. The paper begins with a weight importance analysis, providing an insightful empirical study as a foundation.
2. The proposed method is straightforward to implement.

**Weaknesses:**

Weaknesses:

1. Section 3: The method depends on back-propagation, which could lead to computational and memory overhead challenges in resource-limited scenarios, such as quantizing an extremely large model with only 4 GPUs. However, this issue is acknowledged in Section 8.

2. Section 4: The Fisher information matrix is recalculated at each step. Given that parameters are updated frequently, this could cause the parameter importance to fluctuate significantly, potentially leading to a chain error. This is because gradients influence cherry parameter identification, which in turn affects gradient calculations. The paper does not address how to manage this cyclic error effectively.

3. Section 5: The model for assessing weight importance is based on Optimal Brain Damage and only considers the impact of individual weights when perturbed or quantized. It overlooks the synergistic effects between weights, where quantizing two normal parameters together could have a more significant impact than quantizing one cherry parameter.

4. Section 5: There seems to be a discrepancy in how weight importance is presented: Figure 2 is per weight matrix, whereas Figure 1 is per weight scalar. In the experiments, parameter impact scores are averaged along rows, and cherry parameter identification is performed along columns based on these averages, leading to confusion. If cherry parameter identification is indeed column-based, some cherry parameters may be overshadowed by dominant normal parameters in the same column, leading to potential misidentification.

5. Section 6.2: The improvement in perplexity over OmniQuant is marginal, as shown in Tables 1 and 2. Furthermore, it is weird that OmniQuant is omitted in the QA evaluations in Tables 3 and 4.

I would be inclined to improve the rating if these points are properly addressed.

**Questions:**

Please see Weaknesses.

**Limitations:**

Please see Weaknesses.

---

> ### Author Rebuttal · Authors · 2024-08-07
>
> **W1**: The method depends on back-propagation, which could lead to computational and memory overhead challenges in resource-limited scenarios.
>
> **Response**: While the quantization indeed requires higher computational resources, this is a one-time operation. Once performed, the quantized model will offer significant reduction in computational requirements for the subsequent inference.
>
> **W2**: Section 4: The Fisher information matrix is recalculated at each step. Given that parameters are updated frequently, this could cause the parameter importance to fluctuate significantly, potentially leading to a chain error.
>
> **Response**: We **do not** need to dynamically update the Fisher information matrix results or re-identify cherry parameters. Instead, we only identify them once during the initialization phase (line 1 Algorithm 1). This is because our objective is to ensure each quantization step brings the model closer to the **initial model**, rather than to the model from the previous step. Consequently, the measurement of parameter impact should always be calculated with respect to the initial model, eliminating the need for dynamic updates.
> To experimentally validate this, we conducted an additional experiment comparing two settings:
> 1. (One-time identification) Calculating the Fisher information matrix and identifying cherry parameters only during initialization.
> 2. (Dynamic identification) Recalculating the Fisher information matrix and re-identifying cherry parameters every 40 steps based on the current model state. The perplexities for LLaMA2-7b-4bit-g128 are presented in the table below.
>
> | model| c4 | wiki2|
> |---|:---:|:---:|
> |One-time|7.07|5.58|
> |Dynamic|7.75|6.25|
>
> As evident from the results, dynamically identifying cherry parameters leads to a noticeable decrease in performance. This aligns with our intuition and theoretical understanding.
>
> **W3**: Section 5: The model for assessing weight importance is based on Optimal Brain Damage and only considers the impact of individual weights. It overlooks the synergistic effects between weights.
>
> **Response**: To enhance computational efficiency, we indeed employed the OBD assumption, measuring parameter impacts independently as in Eq (1). To validate this approach, we compared the average impact of two cases: (1) individual parameters ($avg(|H_{ii}|)$), (2) parameter pairs ($avg(|H_{ij}|)$) for LLaMA2-7B and LLaMA2-13B models, sampling 400k values for each case.
>
> | model | $avg(\|H_{ii}\|)$ | $avg(\|H_{ij}\|)$ |
> |-------|:-----------------:|:-----------------:|
> | LLaMA2-7b | 0.0113 | 0.0008 |
> | LLaMA2-13b | 0.0091 | 0.0001 |
>
> This finding indicates that while synergistic effects between weights exist, they are generally much smaller than individual parameter effects.
>
>
> **W4.1**: Section 5: There seems to be a discrepancy in how weight importance is presented: Figure 2 is per weight matrix, whereas Figure 1 is per weight scalar.
>
> **Response**: Figure 1 presents distributions for specific layers, providing more **direct** insights. Figure 2 shows matrix-level distributions, offering a more **global** perspective to ensure that the phenomenon is not biased by individual layers. Both figures demonstrate the existence of parameter heterogeneity.
>
> **W4.2**: In the experiments, parameter impact scores are averaged along rows, and cherry parameter identification is performed along columns based on these averages, leading to confusion. If cherry parameter identification is indeed column-based, some cherry parameters may be overshadowed by dominant normal parameters in the same column, leading to potential misidentification.
>
> **Response**: Parameter impact calculation is **always performed column-wise**. As stated in lines 158-159, "_we average the impact across all rows for each column._" This means we calculate the average impact for each column by considering all rows within that column, not by row-wise calculation.
>
> We also experimented with independently calculating and identifying each cherry parameter, as compared to the column-based approach. The effect is minimal. For instance, with LLaMA2-7B-3bit-g64 on WikiText-2, the separate approach achieved a perplexity of 5.20, only slightly outperform the column-based calculation resulted (5.21). To optimize memory usage and enhance inference efficiency, we opted for the column-based calculation method.
>
> A potential explanation for this phenomenon lies in the low-rank nature of Transformer parameter gradients [1].
>
> [1] GaLore: Memory-Efficient LLM Training by Gradient Low-Rank Projection
>
> **W5**: The improvement in perplexity over OmniQuant is marginal, as shown in Tables 1 and 2. It is weird that OmniQuant is omitted in the QA evaluations in Tables 3 and 4.
>
> **Response**: We have added a comparison on downstream tasks with OmniQuant, as shown in the table below. We use the setting of LLaMA2-7b-3bit-g64.
>
> | model |  Winogrande | ARC-C | GSM8K |
> |---|:---:|:---:|:---|
> |omniquant|71.7|50.0|7.7|
> |CherryQ|71.8|50.6|10.4|
>
> It's important to note that even small improvements in perplexity can translate to significant enhancements in real-world task performance (Table 1 and 2). The additional downstream task comparisons show that CherryQ achieves better results than OmniQuant across various tasks.

---

> > ### Comment · Reviewer_P3Ag · 2024-08-08
> > **.**
> >
> > W1: addressed.
> >
> > W2: Okay.
> >
> > W3: I am not fully convinced. $|H_{ii}| \gg |H_{ij}|$ does not necessarily indicates that the synergy between neurons is much less significant than individual parameter's effect. the $H_{ij}$ are average acrossed pairs of parameters. It is to note that sometimes parameters coalate in larger groups. But I think this problem might be analyzed in a seperate work. Finally, I hope the reviewers cand discuss this point in the limitations.
> >
> > W4: Okay. I carefully read line 158-159 and also said that the importance are averaged across rows. I was asking because Figure 2 was confusing after I read line 158-159. It is better that the authors add some additional explanations in the caption of Figure 2.
> >
> > W5: Okay, athough the average QA performance gain in the above table is only 1.13%.
> >
> >
> > I increase my ratings now. I hope the authors can incorporate my suggestions in their revision. Cheers.

---

> > > ### Author Response · Authors · 2024-08-13
> > >
> > > Thank you for acknowledging our responses and for raising your ratings. We appreciate your valuable feedback and will incorporate your suggestions into the revised version.

---

### Official Review · Reviewer_GdR4 · 2024-07-11

**Soundness:** 2
**Presentation:** 2
**Contribution:** 2
**Rating:** 5
**Confidence:** 3

**Summary:**

This paper investigates the phenomenon of parameter heterogeneity in large language models (LLMs) and proposes a novel quantization method called CherryQ. The authors find that a small subset of parameters, referred to as "cherry" parameters, have a disproportionately large influence on model performance, while the majority of parameters have minimal impact. Based on this observation, CherryQ selectively preserves the critical cherry parameters in high precision while aggressively quantizing the remaining parameters to low precision. Extensive experiments demonstrate that CherryQ outperforms existing quantization approaches in terms of perplexity and downstream task performance.

**Strengths:**

(1) The paper addresses an important issue in large language models and provides valuable insights into the parameter heterogeneity phenomenon.

(2) The experimental results are extensive and well-structured, demonstrating the effectiveness of CherryQ in various settings and benchmarks.

**Weaknesses:**

(1) One major concern is about the implementation. What is *real* improvement in terms of efficiency? The authors are supposed to report the models' throughput and speed compared with baseline models.

(2) The idea of important parameters for quantization is not new and novel.  This paper seems to propose a new term "Cherry Parameters" which appears to be a reinvention of an already existing idea. But the authors could argue the difference with previous studies. I am open to adjust ratings.

**Questions:**

N/A

---

> ### Author Rebuttal · Authors · 2024-08-07
>
> **W1**: What is real improvement in terms of efficiency?
>
> **Response**: We have now conducted real-world tests on a single A100 GPU to measure the generation speed and memory consumption of CherryQ when generating 128 tokens. The results for LLaMA2-7b are summarized in the table below. Compared to FP16, CherryQ at lower precision demonstrates significant improvements in both efficiency and memory usage.
> | model | #tokens/s ($\uparrow$) | peak memory ($\downarrow$) |
> |---|:---:|:---:|
> |FP16| 46.6 | 25843 (MB) |
> |CherryQ-4bit| 68.0 |4552 (MB)|
> |CherryQ-3bit| 71.4 | 3690 (MB) |
>
>
> **W2**: The idea of important parameters for quantization is not new.
>
> **Response**: We appreciate the reviewer's perspective, but respectfully disagree that our idea is not new. Our paper makes several novel and significant contributions:
> 1. **Systematic Revelation of Parameter Heterogeneity**: We are the first to comprehensively demonstrate and analyze the phenomenon of parameter heterogeneity across different LLM scales, families, and types (base and chat models). This fundamental insight provides a unifying explanation for several observed behaviors of LLMs under quantization.
> 2. **Mechanistic Explanation**: We provide the explanation for why LLMs are highly robust to quantization. Our work reveals that this robustness stems from the 99% of parameters that have minimal perturbative impact on model performance.
> 3. **Foundation for Mixed-Precision**: While mixed-precision strategies exist, our work provides the underpinning for why they are effective. We show that these strategies work by preserving the critical "cherry" parameters that have outsized influence on model performance.
> 4. **Novel Metric for Parameter Importance**: Our impact-based metric for identifying important parameters demonstrably superior to existing metrics. We provide both theoretical justification (through our heterogeneity score) and empirical evidence for its effectiveness.
> 5. **Innovative End-to-End Optimization**: CherryQ introduces a novel approach to simultaneously optimize high-precision cherry parameters and low-precision normal parameters - a challenge that existing methods struggle with. Our 3-bit Vicuna-1.5 model matches the performance of 16-bit models, representing a substantial advancement in LLM quantization.
>
> These contributions are outlined in the original manuscript (lines 64-72).

---

### Official Review · Reviewer_YuKp · 2024-07-12

**Soundness:** 3
**Presentation:** 4
**Contribution:** 3
**Rating:** 7
**Confidence:** 4

**Summary:**

This paper presents a new post-training quantization strategy called "CherryQ" that identifies "cherries" --- meaning parameters that disproportionately impact the model's loss --- stores those in fp16, and quantizes the rest of the model parameters using a standard symmetric min-max quantization (with groups).  It shows that this approach yields better perplexity and downstream performance numbers than existing quantization methods, on several models (Llama-3 7b/13, Vicuna 1.5 7b/13b), several tasks (c4/wiki perplexity, Hellaswag, Winogrande, ARC, TruthfulQA, GSM8K, MMLU, chat win-rate), and at several compression rates (2 bits, 3 bits, 4 bits).

Some more details:
- To identify cherries, the paper proposes using E[g_i^2] as a measure of the impact of parameter w_i on the loss, where g_i is the gradient of the loss function with respect to that parameter. This is motivated by: (1) approximating the loss around each parameter w_i using a second order Taylor expansion, assuming E[g_i] = 0 (because model has converged), and assuming the second-order derivative of L w.r.t. w_i is equal to E[g_i^2] (based on results about Fisher information matrix equaling the Hessian from [16]).
- The paper shows that selecting cherries in this way (using the above approximation of the impact of each parameter on the loss) is better than using alternative metrics based on model weights or activations.
- Once cherries are identified, the non-cherries are quantized, and then the model is fine-tuned by training the cherries in a regular manner (non-quantized gradients and updates), while training the (quantized) non-cherries using the straight-through estimator.

**Strengths:**

- The paper demonstrates that some parameters have *much* larger impact on loss than others (at least according to their way of approximating impact), a surprising result, that nicely motivates using mixed precision quantization.
- The proposed method perform well across numerous models, tasks, and compression rates relative to baselines.
- The proposed method is pretty simple and straightforward.

**Weaknesses:**

- It would have been useful to **directly** confirm that the proposed way of approximating impact on loss is actually a good approximation of the impact of a parameter on the loss. This was only done in indirect ways (e.g., by showing that quantizing these parameters leads to better performance than quantizing using other metrics).  It should be relatively straightforward to do this by selecting a random set of model parameters (both cherries and non-cherries), perturbing them slightly (one-by-one), and passing data through the perturbed and unperturbed model and measuring the loss difference.  One could then show a scatter plot (and measure spearman rank correlation) to visualize how well the approximated impact on loss (E[g_i^2]) aligns with the actual impact on loss.
- QuIP (https://arxiv.org/abs/2307.13304) and QuIP# (https://arxiv.org/abs/2402.04396) are not compared to, and more recent state-of-the-art models are not benchmarked (e.g., Llama3).
- No inference speed results are shared. It would be great to see if this method can directly lead to faster inference, especially during decoding (which is memory-bound).

**Questions:**

- See weaknesses. Are there any obstacles to running the experiment to validate the use of E[g_i^2] as the metric for measuring impact on loss?
- How "data-dependent" are the cherries? Can this be measured?
- How does CherryQ compare with QuIP/QuIP# on Llama-3?

**Limitations:**

Limitations are discussed, though could be addressed better with some experiments (see questions above).

---

> ### Author Rebuttal · Authors · 2024-08-07
>
> **W1 and Q1**: It would have been useful to **directly** confirm that the proposed way of approximating impact on loss is actually a good approximation of the impact of a parameter on the loss.
>
> **Response**: Following your suggestion, to directly validate the accuracy of our impact approximation, we investigated the effects of different types of parameter perturbations on the model's loss. We compared the following two settings for LLaMA2-7b:
> 1. Adding perturbations in the range of (-0.5, 0.5) to all cherry parameters.
> 2. Randomly sampling an equal number of normal parameters and adding perturbations in the range of (-0.5, 0.5) to them.
>
> We repeated this random sampling and perturbation process five times and measured their perplexities. The results are presented in the table below. Perturbations to cherry parameters led to significantly larger increases in perplexity, while normal parameters remained largely insensitive to similar perturbations. This directly validates both the existence of cherry parameters and the effectiveness of our approximation.
>
> |model | c4 | wiki2 |
> |--------|:---:|:---:|
> |LLaMA2-7b | 6.97 | 5.47 |
> |Perturb Normal Parameters|6.98($\pm$ 0.00) | 5.48($\pm$ 0.00)|
> |Perturb Cherry Parameters|8.12($\pm$ 1.78) | 6.12($\pm$ 0.84)|
>
> **W2 and Q3**: How does CherryQ compare with QuIP/QuIP# on Llama-3?
>
> **Response**: We have compared CherryQ with QuIP and QuIP# on LLaMA2-7b in terms of perplexity, as shown in the table below. We haven't included comparisons over LLaMA-3 as we couldn't find QuIP/QuIP# implementations based on that model.
>
> | Model | Bits | Wiki | C4 |
> |-------|------|------|----|
> | 7b-fp16 | 16 | 5.47 | 6.97 |
> | 7b-QuIP | 4 | 5.94 | 8.01 |
> | 7b-QuIP# | 4 | 5.56 | 7.07 |
> | 7b-CherryQ | 4 | 5.58 | 7.07 |
> | 7b-QuIP | 3 | 6.50 | 8.74 |
> | 7b-QuIP# | 3 | 5.79 | 7.32 |
> | 7b-CherryQ | 3 | 5.93 | 7.39 |
> | 13b-fp16 | 16 | 4.88 | 6.47 |
> | 13b-QuIP | 4 | 5.01 | 6.89 |
> | 13b-QuIP# | 4 | 4.95 | 6.54 |
> | 13b-CherryQ | 4 | 4.99 | 6.56 |
> | 13b-QuIP | 3 | 5.34 | 7.34 |
> | 13b-QuIP# | 3 | 5.10 | 6.72 |
> | 13b-CherryQ | 3 | 5.26 | 6.80 |
>
> CherryQ shows clear improvements over QuIP under the same settings. While QuIP# achieves better results, its main advantage stems from vector quantization, a technique  primarily addressing how to replace the quantization of individual weights with quantization for groups of weights. This technique is orthogonal and complementary to our approach rather than directly comparable.
>
> **W3**: No inference speed results are shared.
>
> **Response**: We have now conducted real-world tests on a single A100 GPU to measure the generation speed and memory consumption of CherryQ when generating 128 tokens. The results for LLaMA2-7b are summarized in the table below. Compared to FP16, CherryQ at lower precision demonstrates significant improvements in both efficiency and memory usage.
> | model | #tokens/s ($\uparrow$) | peak memory ($\downarrow$) |
> |---|:---:|:---:|
> |FP16| 46.6 | 25843 (MB) |
> |CherryQ-4bit| 68.0 |4552 (MB)|
> |CherryQ-3bit| 71.4 | 3690 (MB) |
>
> **Q2**: How "data-dependent" are the cherries? Can this be measured?
>
> **Response**: To validate the data dependence, we conducted additional experiments in terms of data selection and data scale:
>
>  - **Data Selection Independence**: We validated that cherry parameters identified using different data samples are largely consistent. Specifically, we randomly selected **five sets of 128 samples** from WikiText-2 and calculated cherry parameters for each set. We found an **84%** overlap rate on average between any two different cherry parameter sets, indicating that different data selections tend to yield similar cherry parameters.
>  - **Data Scale Independence**: We also verified that cherry parameters remain consistent across different data scales. We randomly selected four sample sets from WikiText-2 with **sizes of 128, 256, 512, and 1024**, respectively, and calculated their corresponding cherry parameters. The average overlap rate of cherry parameters between any two sets reached **89%**, demonstrating that varying data scales tend to produce similar cherry parameters.
>
> These findings demonstrate that cherry parameters can be reliably identified using a relatively small set of samples (as few as 128), enhancing both the generalizability and efficiency of our method.

---

> > ### Comment · Reviewer_YuKp · 2024-08-13
> >
> > Thank you for your thorough answers!
> >
> > Quick follow-up question for Q2: Are cherry parameters consistent across datasets (e.g., C4 and wikitext)?

---

> > > ### Author Response · Authors · 2024-08-14
> > >
> > > We conducted a cross-dataset analysis to evaluate the consistency of cherry parameters across different datasets. Specifically, we compared cherry parameters identified from C4 and WikiText datasets. We performed 9 independent sampling trials, each using 128 samples from both datasets.
> > >
> > > Our analysis revealed an **average overlap rate of 68%** between the cherry parameters identified from C4 and WikiText. This overlap, while lower than the 84% observed within the same dataset, still represents a substantial consistency across datasets. The difference can be attributed to dataset-specific characteristics influencing parameter importance. This suggests a strong core consistency in cherry parameters. There are also potentials for dataset-specific optimizations in future research.

---

### Decision · Program_Chairs · 2024-09-25

**Decision:**

Accept (poster)

**Comment:**

This paper can be summarized, as reviewer YuKp says, "The paper demonstrates that some parameters have much larger impact on loss than others (at least according to their way of approximating impact), a surprising result, that nicely motivates using mixed precision quantization." And regarding execution: as reviewer GdR4 says, "The experimental results are extensive and well-structured, demonstrating the effectiveness of CherryQ in various settings and benchmarks."

Overall ratings were positive, and there were relatively minor revisions required during the rebuttal process. And what revisions were made, made the paper stronger in a quick and understandable way. For instance, there was a direct validation of impact approximation. Reviewer YuKp: "It would have been useful to directly confirm that the proposed way of approximating impact on loss is actually a good approximation of the impact of a parameter on the loss."
To which the authors responded with an experiment resulting in: "Perturbations to cherry parameters led to significantly larger increases in perplexity, while normal parameters remained largely insensitive to similar perturbations. This directly validates both the existence of cherry parameters and the effectiveness of our approximation." Other concerns about cherry parameter dataset overlap and memory-bound latency were also addressed to GdR4 and YuKp.

There are some limitations though with the paper. P3Ag is still not convinced that |Hii| >> |Hij|. And they also point out that the average QA performance gain over OmniQuant is still relatively small at 1.13%.

Overall, the paper makes a significant contribution to the field of LLM quantization through its novel insights and effective method. The strong author responses and extensive evaluation support its acceptance.